# Genome-wide excision repair in *Arabidopsis* is coupled to transcription and reflects circadian gene expression patterns

Onur Oztas[1], Christopher P. Selby[1], Aziz Sancar[1] & Ogun Adebali[1]

Plants are exposed to numerous DNA-damaging stresses including the exposure to ultra-violet (UV) component of solar radiation. They employ nucleotide excision repair to remove DNA-bulky adducts and to help eliminate UV-induced DNA lesions, so as to maintain their genome integrity and their fitness. Here, we generated genome-wide single-nucleotide resolution excision repair maps of UV-induced DNA damage in *Arabidopsis* at different circadian time points. Our data show that the repair of UV lesions for a large fraction of the genome is controlled by the joint actions of the circadian clock and transcription by RNA polymerase II. Our findings reveal very strong repair preference for the transcribed strands of active genes in *Arabidopsis*, and 10–30% of the transcription-coupled repair is circadian time-dependent. This dynamic range in nucleotide excision repair levels throughout the day enables *Arabidopsis* to cope with the bulky DNA lesion-inducing environmental factors including UV.

[1] Department of Biochemistry and Biophysics, University of North Carolina School of Medicine, Chapel Hill, NC 27599-7260, USA. Correspondence and requests for materials should be addressed to A.S. (email: aziz_sancar@med.unc.edu) or to O.A. (email: adebali@unc.edu)

Plants are sessile and rely on photosynthesis to harvest energy. Thereby, they are exposed to a substantial amount of the ultraviolet (UV) component of solar radiation and other environmental stresses throughout the day[1]. This lifestyle inflicts a high level of DNA damage on the plant genome that impairs genome integrity, growth and development[2,3]. Nucleotide excision repair (excision repair) corrects a wide range of bulky DNA adducts and it is the sole mechanism to repair the majority of these lesions[4]. UV-induced DNA lesions, in the form of cyclobutane pyrimidine dimers (CPDs) and (6–4) pyrimidine-pyrimidone photoproducts, are repaired by excision repair and blue light-dependent photoreactivation. Excision repair pathway directly recognizes bulky DNA lesions (global repair) and removes the lesion-containing oligomers by concerted dual (5′ and 3′) incisions, followed by gap filling and ligation[4]. The repair rate is strongly stimulated by the lesion which blocks the transcription by RNA polymerase II (transcription-coupled repair, TCR).

Excision repair has been relatively well characterized in mammalian cells, which requires six factors (XPA, RPA, XPC, TFIIH, XPG and XPF-ERCC1) to incise and release the lesion-containing oligomers. In addition to these factors, CSA and CSB proteins are needed to remove DNA lesions by TCR. Mammalian excision repair genes are conserved in *Arabidopsis thaliana* and other plants; however, they lack an apparent *XPA* ortholog[5,6]. Nevertheless, in addition to photoreactivation, *Arabidopsis* has been shown to perform mammalian-type excision repair to remove UV-induced DNA lesions[7–10]. Furthermore, plants defective in excision repair exhibit UV hypersensitivity[2,11,12]. In the absence of excision repair, plant genomes accumulate mutations even in a UV-free condition[2], which implies that excision repair has a broader range of substrates in plants, as in all other organisms investigated. Therefore, excision repair is vital in maintaining genome integrity and plant fitness against the bulky adducts induced by a variety of sources.

Although excision repair has been identified in plants, the profile and dynamics of this repair mechanism throughout the genome and its regulation remain poorly understood. Herein, we generated the genome-wide excision repair map of CPDs at single-nucleotide resolution and investigated roles of global regulatory mechanisms on excision repair in *Arabidopsis*. We find that transcription is the major factor determining the excision repair profile of transcribed strand (TS) of expressed genes throughout the genome. TCR exhibits circadian rhythmicity in 10–30% of total genes. The synchronized repair rhythmicity in clusters of genes coordinates the repair of biological pathways throughout the day. Our study monitored the genome-wide dynamics of nucleotide excision repair which is an important mechanism that plants use to cope with bulky lesion-inducing environmental factors, specifically the UV component of solar radiation.

## Results

**Excision repair map of CPDs at single-nucleotide resolution.** We used the excision repair-sequencing (XR-seq) method to create genome-wide excision repair maps of CPD damage. Briefly, we isolated the CPD-containing oligonucleotides removed by excision repair and amplified them to generate libraries. Then, we sequenced the libraries and aligned the reads to *Arabidopsis* genome[13,14]. We applied XR-seq to *Arabidopsis* seedlings that were harvested 30 min after UV treatment (Supplementary Fig. 1). This is a relatively early time point in the CPD repair time course which takes hours to complete[10]. To prevent

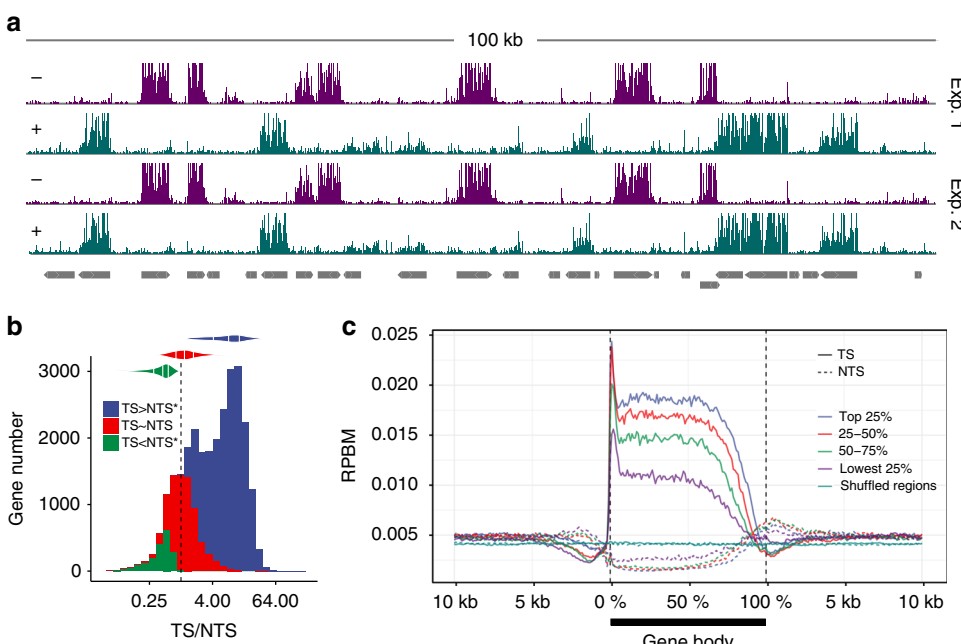

**Fig. 1** Prevalence of transcription-coupled repair throughout the genome. **a** Strand-specific XR-seq signals from two biological replicates illustrating the excision repair in the 100 kb region starting at 15,392 kb of chromosome 5. Purple and green represent plus (5′ to 3′—left to right) and minus (5′ to 3′—right to left) strands, respectively. Gray horizontal bars show the genes each of which has an arrow indicating its direction. **b** Distribution of TS/NTS repair ratios of the annotated protein-coding genes (~27,000) in histogram and violin plot format. Untransformed ratio values are shown on the log2-scaled x-axis. The vertical dashed line (x = 1) is where TS repair is equal to NTS. Blue (TS > NTS) and green (TS < NTS) represent significant asymmetrical repair between two strands (FDR < 0.05) and red shows the remaining genes. **c** Repair profiles of the transcribed and flanking regions of the ~9000 expressed genes. Gene body lengths are scaled to percentage and the length of each flanking region is 10 kb divided into bins of 100 bp. The genes were grouped into quartiles based on their transcription levels. Shuffled regions are the randomly repositioned genes by keeping their lengths constant

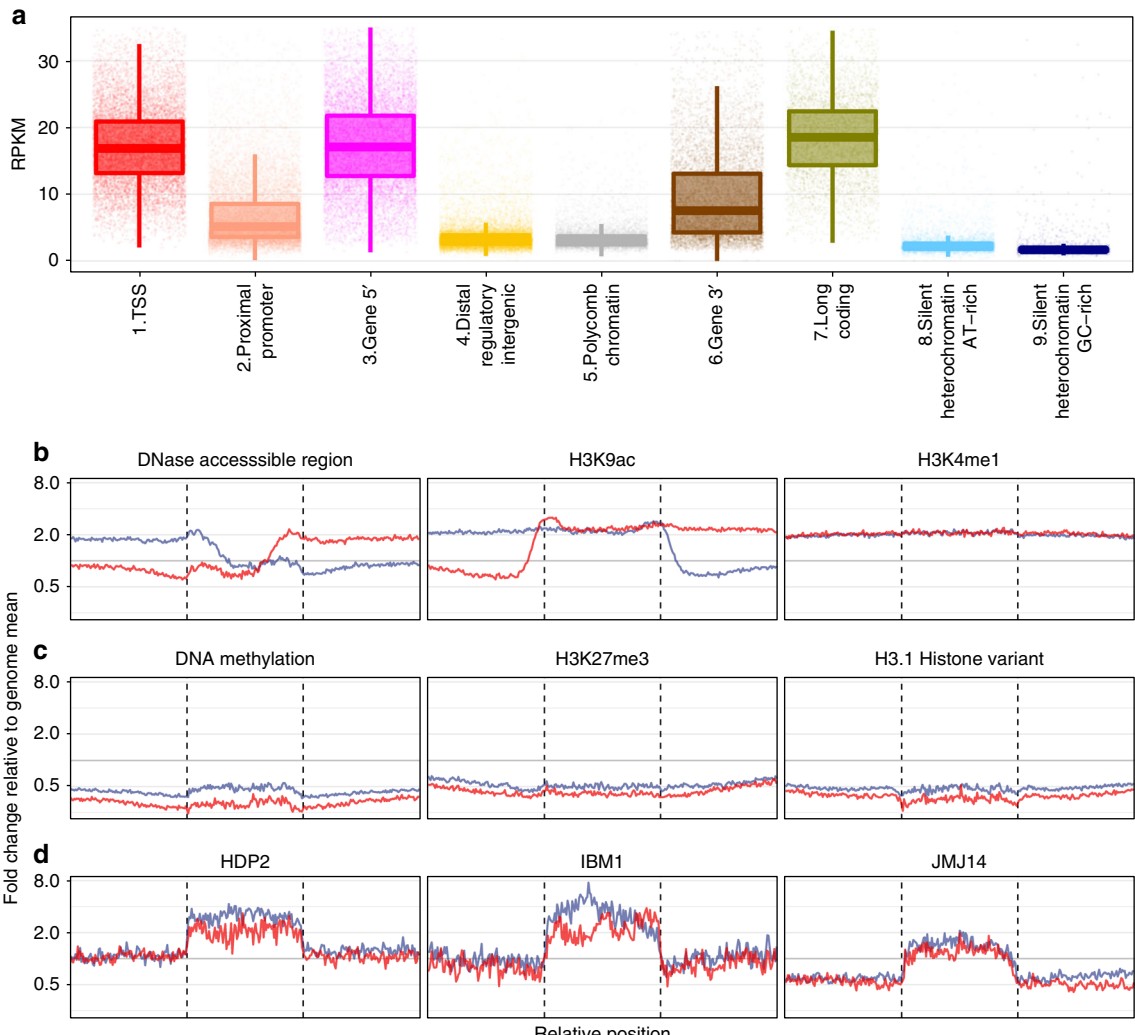

**Fig. 2** Chromatin state influence on genome-wide kinetics of nucleotide excision repair. **a** The level of excision repair for each chromatin state. In the boxplot, the middle line corresponds to median, the lower and upper hinges correspond to first and third quartiles. The whiskers extend from hinges to 1.5*IQR (inter-quartile range, the range between first and third quartiles). The excision repair profiles (**b**), at the sites of DNAse hypersensitivity, H3K9 acetylation, H3K4 monomethylation (**c**), in regions of DNA methylation, H3K27 trimethylation and H3.1 histone variant binding. **d** The change of excision repair level at the binding sites of HDP2, IBM1 and JMJ14. **b–d** The regions between vertical dashed lines are the sites (scaled to percentage) where the factor interacts with DNA and each flanking site is 1 kb. The log2-scaled *y*-axis show untransformed values of fold-change. The horizontal gray line (*y* = 1) is where the signal is equal to the genome mean. Blue and red represent plus and minus strands, respectively

photoreactivation, we kept the seedlings in the dark after UV-irradiation. Consistent with our previous work with the T87 *Arabidopsis* cell line[10], we detected primary excision products 23–27 nts in length, as shown with excision assay results (Supplementary Fig. 2), and with XR-seq results (Supplementary Fig. 3) illustrating frequency distribution of excision product lengths associated with plants irradiated at several circadian time points (ZT2-ZT23, discussed below). The primary excision products are produced by incisions 4–6 nt 3′ and 18–21 nt 5′ from the adduct[10], which is indicated by the frequency of nucleotides at each position of a 27mer excision product (Supplementary Fig. 4a). We also detected a population of fragments 10 to 22 nt in length (Supplementary Fig. 3). Shortening of excision products from 24–32 to 10–20 nt is associated with the loss of nucleotides from the 5′ end, which we inferred from the thymine enrichment at a fixed position relative to the 3′-end in excision products 11 to 32 nt in length (Supplementary Fig. 4b). This loss is consistent with the pattern of degradation from the 5′ end observed in other eukaryotes.

**Transcription and excision repair.** The *Arabidopsis* genome has orthologs of mammalian CSA and CSB genes known to be required for TCR. Also, a preliminary study showed TCR in a single *Arabidopsis* gene by demonstrating that the TS is repaired about five-fold more efficiently than the non-transcribed strand (NTS)[15]. We therefore first analyzed the effects of transcription on repair by examining repair asymmetry between strands and found that *Arabidopsis* exhibits an unprecedented preference for the TS over NTS repair in a set of, presumably transcriptionally active, annotated genes, that is consistent across biological replicates (Fig. 1a). For the majority of the annotated protein-coding genes (~27,000) the TS is repaired preferentially (Fig. 1b). The genes showing no strand preference for repair are either non-transcribed or weakly transcribed. Those with TS/NTS <1 are mostly genes with apparent overlapping annotated or unannotated anti-sense transcription (Supplementary Fig. 5). We next determined the repair profiles of the transcribed and flanking regions of the ~9000 expressed genes previously reported to be actively transcribed by the GRO-seq method (Fig. 1c)[16]. We

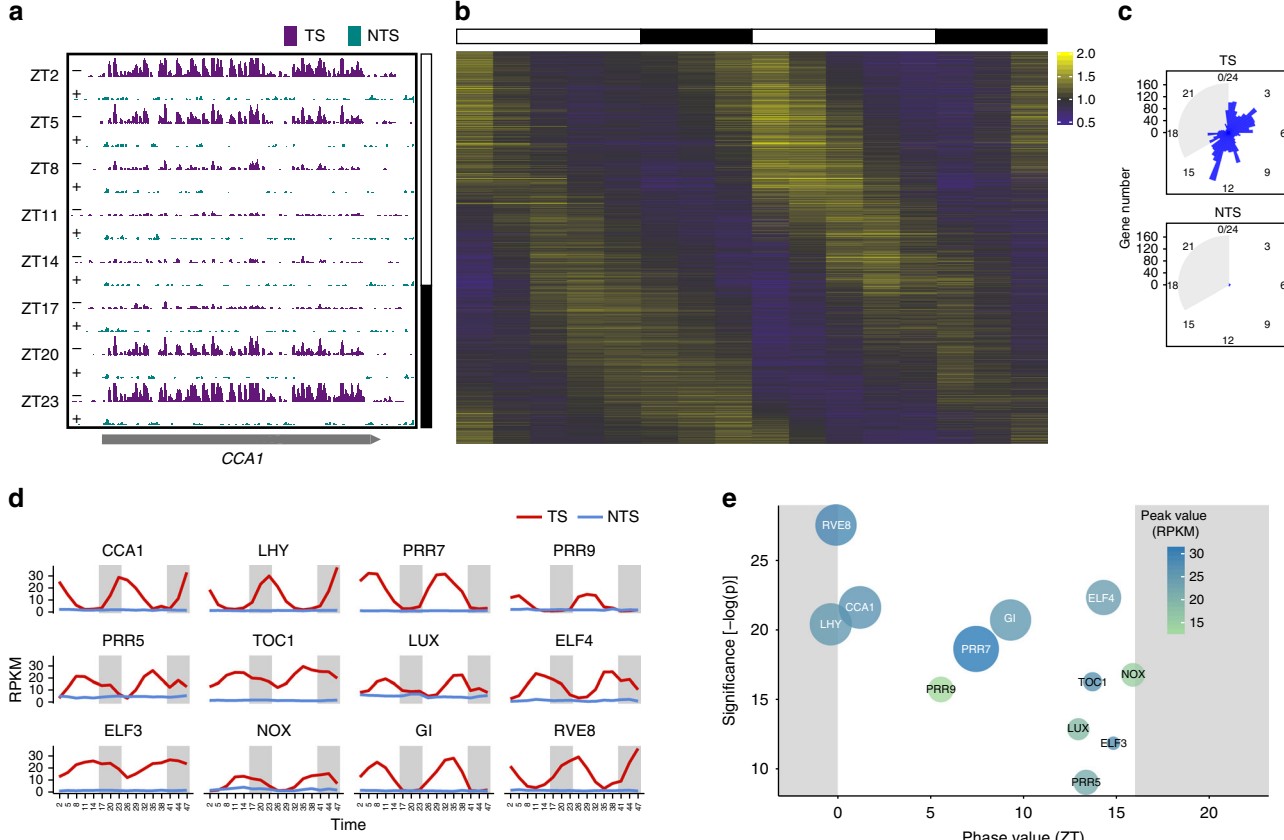

**Fig. 3** Circadian oscillation of transcription-coupled repair. **a** Repair of the of *CCA1* TS (purple) and NTS (green) at different circadian time points. **b** Heat map of the relative repair levels of ~4000 genes showing circadian repair rhythmicity in their transcribed strands. The genes are sorted by their phase values. The value represented is the observed/median repair ratio per gene. **c** Phase value distribution of the genes having an oscillating repair in their TS (top) and NTS (bottom). The plot is a circularized histogram allowing a fluent circadian scale. **d** The circadian repair profiles of the core clock genes. The *y*-axis shows the repair signal (RPKM) whereas *x*-axis represents time of the concatenated two experiments. The gray background represents the 8 h dark period. **e** Phase value (*x*-axis) vs. −log *p* value (significance of rhythmicity computed with Metacycle software, *y*-axis) profile of the core clock genes. Circle sizes indicate peak-to-trough read amplitudes and color gradient indicates peak values

found significant enrichment of repair in TS relative to NTS for each expression quartile. The repair peak near the transcription start site on the TS is potentially caused by abortive transcription[17] rather than proximal promoter RNAPII pausing which is thought to be lacking in *Arabidopsis*[16]. The magnitude of TS repair positively correlates with the level of transcription (Spearman correlation coefficient: 0.46). However, TCR appears to be saturated in the highly expressed genes (Supplementary Fig. 6) which could be due to either a finite number of CPDs or the mechanistic limits of TCR. In addition, it appears that the NTS is repaired less efficiently than the intergenic regions, likely owing to TCR in the flanking regions caused by neighboring genes (Supplementary Fig. 7). The difference in repair activity between the two strands is not the result of sequence content bias. In fact, the overall dithymine (TT) frequency is slightly higher in the NTS (Supplementary Fig. 8). We conclude that TCR dictates the genome-wide excision repair profile in *Arabidopsis* under our experimental conditions.

**Chromatin states and excision repair**. In *Arabidopsis*, nine chromatin states enriched with combinatorial genomic elements have been identified[18]. We computed the repair level in each state to understand the impact of chromatin state on excision repair efficiency. The chromatin state 2 (transcription-start sites), 3 (5′ coding regions) and 4 (long coding regions) are repaired most

efficiently followed by the state with the 3′ coding regions (Fig. 2a). The difference in repair between the 5′ and 3′ regions of the genes likely follows in part from the presence of multiple CPDs in many genes under our conditions (about 0.7/kb). Due to the concentration of RNAPII at the 5′ end of genes, and dissociation of the blocked RNAPII from the template during TCR[19], there is a delay in transcription to lesions at the 3′-end of genes. The chromatin state 2 enriched in promoter proximal regions showed a slightly higher repair efficiency than state 4 with distal regulatory intergenic regions and Polycomb-mediated transcriptionally repressed state 5, consistent with the more open chromatin structure in the promoter proximal regions. We also observed higher repair levels in the non-transcribed open chromatin states 4 and 5 compared to heterochromatic states (8 and 9).

Chromatin status at a locus is influenced by the presence of histone variants, histone posttranslational modifications and DNA methylation[20]. We therefore analyzed the effects of these factors on repair. We observed a higher repair efficiency in DNase accessible regions and at sites with H3K9ac and H3K4me1 marks, which are associated with open chromatin (Fig. 2b). The asymmetrical repair profiles of DNase accessible and H3K9ac-bound sites for each strand are due to the downstream transcribed regions. As expected, DNA methylation, trimethylation of H3K26 and the presence of H3.1 histone variant caused severe depression in the repair of both strands because these

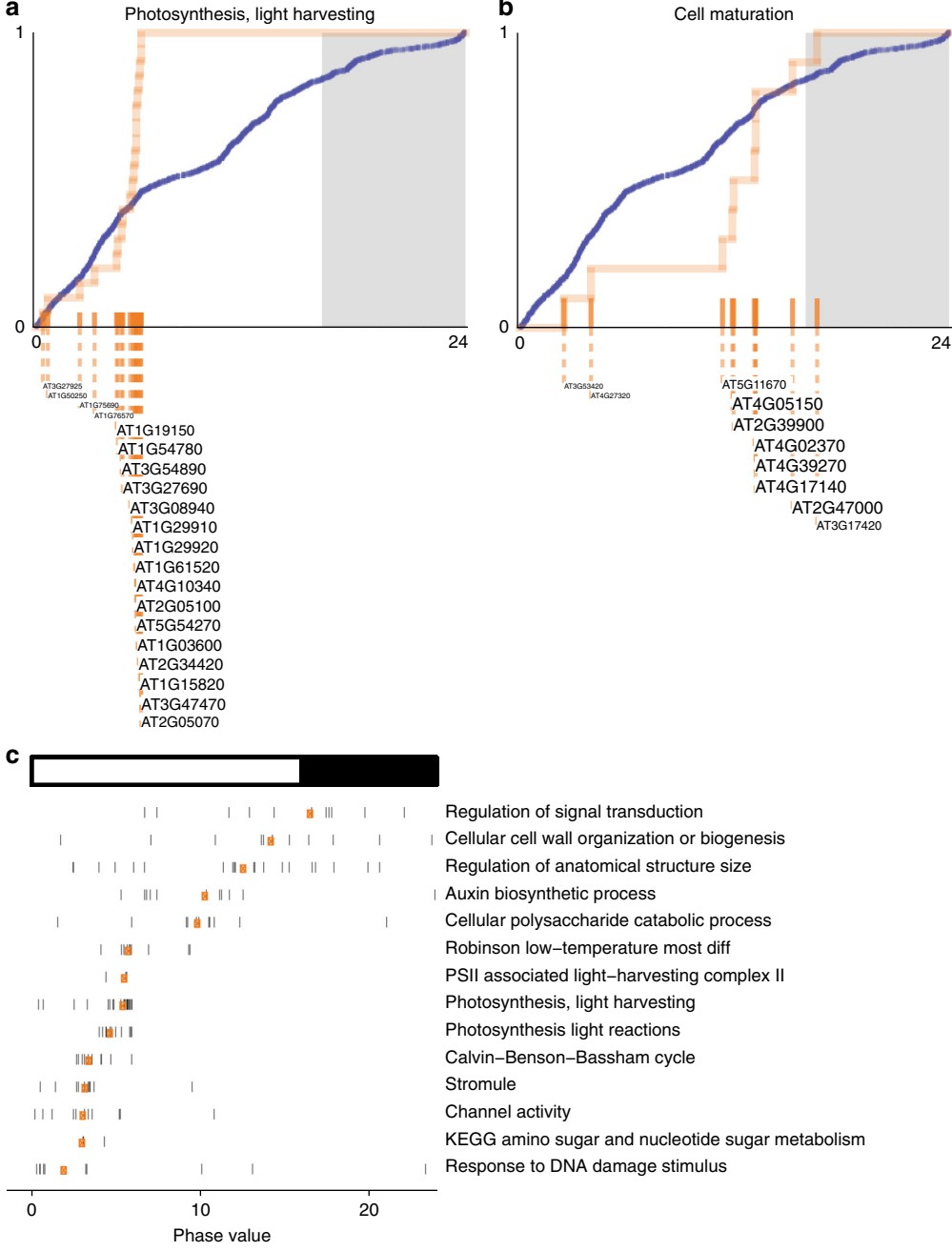

**Fig. 4** Phase set enrichment analysis of the genes with oscillating TCR. Cumulative distributions (to 100%) of the genes associated with **a** Photosynthesis, light harvesting and **b** Cell maturation categories (orange). The cumulative distributions are significantly different from the background (1453 genes with TCR oscillation, blue) based on the Kuiper test (*q* < 0.05). **c** Phase values of the genes involved in specific pathways. Each vertical bar represents a gene phase value which is aligned with the *x*-axis. Orange points represent the median phase values of the genes for each category

marks are associated with heterochromatin (Fig. 2c). The slight preference of repair in the minus strand is, in fact, a reflection of the asymmetrical TCR because of the uneven distribution of gene orientations on two strands. Finally, the binding of HDP2, an anti-silencing factor of DNA methylation[21] and histone demethylases IBM1[22] and JMJ14[23] promoted the repair of both strands compared to the neighboring regions which are repaired at or below the average genome-wide repair level (Fig. 2d). By using chromatin databases derived from nonidentical growth regimes and developmental time points in our analysis, we reached the same conclusion that the repair level in euchromatic regions is favored compared to heterochromatic regions. Although chromatin landscape depends on the experimental conditions, the cumulative effects of each state (thousands of genomic locations) on repair were captured consistently. To sum up, our results confirmed the expectations that the euchromatic and transcribed regions are repaired more efficiently than heterochromatic regions.

**Circadian clock and excision repair.** *Arabidopsis* possesses a rather sophisticated circadian clock in the form of a transcription–translation feedback loop that controls all major biochemical pathways including nearly 30% of all *Arabidopsis* genes[24–26]. Because plants are exposed to UV damage on a diurnal cycle, we hypothesized that excision repair may be influenced by circadian clock regulation. To study the interface of

the circadian clock with excision repair, we analyzed the repair of *Arabidopsis* circadian genes as well as of the entire genome over a circadian cycle by performing XR-seq at 3-h intervals in a long day condition (ZT2-ZT23). We first examined the repair pattern of one of the key circadian clock genes, *CCA1*, a so-called "morning gene" known to be expressed at pre-dawn/dawn hours (Fig. 3a). The repair pattern of *CCA1* exhibits a dramatic oscillatory pattern of TS repair, in which the zenith (ZT23)-to-nadir (ZT11) ratio (amplitude) is 15. The NTS repair shows no circadian oscillation. Analysis of the entire genome over a circadian cycle showed that ~5% of the genome is repaired in a rhythmic manner (Supplementary Fig. 9). Because TCR is a major component of *Arabidopsis* DNA repair, we distinctly analyzed TS and NTS repair of each gene. The results showed that ~4000 genes exhibit a circadian pattern of repair in the TS with maxima phases spread over the entire cycle (Fig. 3b). The computed phase value distribution of genes with an oscillating pattern of TS repair shows two main phases of maximum repair at dawn (ZT0 to 4) and dusk (ZT12–16) (Fig. 3c). Although the number of genes with oscillating repair in their NTS is minor, they exhibit the general dawn/dusk type of rhythmicity (Fig. 3c). The minor NTS repair oscillation is likely due to TCR of overlapping genes with an opposite orientation, or presumably unannotated anti-sense transcription (Supplementary Fig. 10). Finally, the core circadian clock genes vary widely in their TS repair levels (Fig. 3d) and their repair maxima phases. Most importantly, the repair maximum phase of each clock gene coincides with its reported transcription maximum phase[27] (Fig. 3e, Supplementary Fig. 11). As with other genes, the NTS repair is very low among core clock genes over the entire cycle. To conclude, the circadian clock profoundly influences TCR via rhythmic gene expression.

**Coordinated repair of pathways**. The circadian clock coordinates the expression of components of biochemical and signaling pathways in *Arabidopsis*[28]. To investigate the possibility of coordinated repair within these pathways, we performed functional enrichment analysis of the phase values for the genes with TCR oscillation. We found that specific genes involved in the same pathway are repaired most efficiently within a narrow temporal window. For example, while the TCR of photosynthesis-associated genes peaked between ZT0 and ZT6 (Fig. 4a), the genes of cell maturation exhibited TCR maxima between ZT12 and ZT16 (Fig. 4b). Numerous pathways showed a repair peak at a different time of day based on the TCR oscillation of the component genes (Fig. 4c, Supplementary Fig. 12). While these oscillations demonstrate circadian clock control of gene expression, in some cases, such as light-responsive photosynthetic genes, cyclic expression is controlled directly by diurnal environmental exposures. The net effect is to enable plants to efficiently use their repair capacity to maintain biological processes throughout a day. Overall, our analysis demonstrates that plants temporally orchestrate the repair of key pathways using TCR.

## Discussion

As sessile organisms, plants cope with bulky DNA lesion-inducing environmental factors. They possess an excision repair mechanism to remove these lesions to ensure their genomic stability. In this study, we focused on the CPD photoproduct, which is the most abundant bulky DNA lesion caused by the UV-component of solar radiation. We monitored the genome-wide dynamics of plant excision repair by generating repair maps of CPDs for the *Arabidopsis* genome over an entire circadian period and showed transcription and circadian rhythms together create a high dynamic range in repair across a large fraction of the genome.

We investigated the effects of three regulatory mechanisms on plant excision repair: transcription, the circadian clock and chromatin state. Our data revealed that the main determinant of genome-wide excision repair profile is transcription. This transcription-driven repair exhibits circadian rhythmicity in up to 30% of genes; however, global repair does not oscillate. In mammalian cells, it was reported that global repair does oscillate, and peaks at ZT10 [29,30]. This oscillation results from the circadian rhythmicity of XPA expression, implying that the lack of global repair oscillation in plants is due to the absence of an *XPA* ortholog. Our analysis also showed that epigenetic factors influence excision repair: the repair level in euchromatic regions is higher than heterochromatic regions due to the difference in the accessibility of repair factors to damage sites. Our result is consistent with the observation that UV causes higher mutation rates of DNA methylated cytosines in heterochromatic regions than euchromatic regions[2].

Plant circadian studies utilizing microarray and RNA-seq methods capture the reflection of not only transcriptional but also posttranscriptional events that affect the eventual levels of mature RNA. XR-seq circumvents posttranscriptional regulation by monitoring transcription owing to the strong TCR. Our data can be used to distinguish the effects of these two regulatory systems on plant circadian clock. In general, oscillation patterns of well-characterized genes obtained by XR-seq are consistent with the earlier circadian RNA expression data[25,27]. The difference in the rhythmicity patterns of individual genes, if any, might be due to the direct measurement of transcription process by XR-seq as well as the different experimental conditions and UV irradiation.

UV stress studied here is one of the environmental DNA damaging factors that are reversed by excision repair. The accumulation of mutations in the absence of excision repair even in a UV-free condition suggests that excision repair has a broader range of substrates and thus has additional roles in maintaining genome integrity and plant fitness[2]. All these make excision repair a potential engineering target to improve yield. We believe the findings presented here should be of benefit for plant husbandry, and for crop improvements for staple plants, such as rice and wheat, which appear to have circadian clocks similar to that of *Arabidopsis*.

## Methods

**Plant materials and growth conditions**. Ten-day-old seedlings of *Arabidopsis thaliana* Columbia (Col-0) accession were used. Plants were grown under a long-day condition (16 h light/8 h dark) with a cool white fluorescent light at 24 °C. Eight milligrams of seeds for each sample were surface-sterilized, and stratified for 2 days at 4 °C, and then planted on a Murashige and Skoog plate. For circadian clock experiments, seedlings were collected in 3-h intervals (ZT2, ZT5, ZT8, ZT11, ZT14, ZT17, ZT20, and ZT23).

**Excision assay**. Ten-day-old seedlings were irradiated with 1 J/(m²s) UVC (254 nm) for 2 min (120 J/m² UVC). After 30-min incubation in the dark at 24 °C, the seedlings were frozen with liquid nitrogen, and were ground using mortar and pestle. The resulting powder was resuspended in 400 μl of STES buffer (200 mM Tris·HCl pH 8.0, 500 mM NaCl, 0.1% SDS, 10 mM EDTA) and 400 μl of phenol: chloroform (20:1). The sample was homogenized by vortexing with acid-washed glass beads for 30 min at 4 °C, followed by centrifugation at 14,000 rpm for 10 min at room temperature. The supernatant was treated with 10 μl of RNAseA (R4642; Sigma) for 1 h at 37 °C, then with 10 μl of proteinase K (P8107S; NEB) for 1 h at 60 °C. The excision products were obtained by ethanol precipitation, and purified by immunoprecipitation with a CPD-specific antibody obtained from Cosmo Bio. (NMDND001). These fragments were 3′-end radiolabeled with [α-32P]-3′-deoxyadenosine 5′-triphosphate (cordycepin 5′-triphosphate) (Perkin-Elmer) by terminal deoxynucleotidyl transferase (NEB), and visualized on an 11% sequencing gel.

**XR-seq library preparation**. Excision products were purified as above. 5′ and 3′ adapters compatible with the Illumina TruSeq Small RNA protocol were ligated to excision products. Ligation products were immunoprecipitated with CPD antibodies (CosmoBio USA), then photoreversed with photolyase. Analytical PCR was

performed using one percent of the sample to decide the minimum number of cycles required for preparative scale PCR amplification (Supplementary Figure 2b). The repaired ligation products were PCR-amplified using 50- and 63-nt-long primers adding specific barcodes compatible with the Illumina TruSeq Small RNA kit. The correct size PCR products representing the library were gel purified, and then sequenced in the Illumina HiSeq 4000 (experiment 1) and 2500 (experiment 2) platforms, and single-end 50-nt reads were generated. Two experiments each with 8 samples collected at different circadian time points were performed based on the ENCODE[31] and circadian studies guidelines[32].

**XR-seq data preprocessing**. The reads were processed with cutadapt to trim the adapter sequences (TGGAATTCTCGGGTGCCAAGGAACTCCAGTNNNNNNNA CGATCTCGTATGCCGTCTTCTGCTTG) from the 3′-end. Bowtie[33] was used to align XR-seq reads onto *Arabidopsis* genome (TAIR10) with the following parameters:–nomaqround --phred33-quals -e 70. File conversions were processed with samtools[34] and bedtools[35]. Read duplications were filtered out by keeping the unique genomic regions only.

**Read length distribution and nucleotide frequency**. The read length distributions and nucleotide abundance plots were plotted using the mapped reads by custom scripts and R ggplot package for each sample. Dithymine frequencies were computed using the merged samples (16: experiment 1 and 2 with 8 time points for each) data file.

**Screenshots**. Bed files were converted to bedgraph format by applying RPM (reads per million mapped reads) normalization with the --scale option[35] followed by bigwig conversion using UCSC tools[36]. All screenshots were captured using IGV[37]. Figure 1a screenshot represents the merged samples distinctly for experiment 1 and 2.

**Genic repair levels**. Strand-specific repair level for each gene was computed with bedtools[35] and custom scripts. Gene-body and up/down-stream repair profiles were computed using the bedtools combined with custom scripts. Basically, we divided each gene into 100 bins (each representing 1% of the total gene length) independent of their length for gene body. For the flanking regions, unscaled bin-based counts were plotted. Normalization to RPBM (reads per base per million mapped reads) was applied.

Comparing TS and NTS repair values for each gene was performed by pooling two merged samples (ZT2, 8, 14, 20 and ZT5, 11, 17, 23) from each of the two experiments. Then we applied *t*-test on the four samples and corrected the *p*-values to retrieve an FDR value for each gene. FDR < 0.05 cutoff was used for the significant difference decision between TS and NTS repair.

**Chromatin states and epigenetic markers**. The nine chromatin states were retrieved from Sequeira-Mendes et al.[18]. The annotations for chromatin states were obtained from Vergara and Gutierrez[38]. DNase hypersensitivity sites (SRX11100), H3K9ac (SRX1466723), H3K4me1 (DRX066785), H3K27me3 (SRX648274), H3.1 Histone Variant (SRX113876), IBM (DRX066817), HDP2 (SRX2310525) ChIP-seq data sets and DNA methylation (SRX004968) meDIP data set were retrieved from PCSD database in BigWig formats. Peaks were called by using MACS2 with default parameters[39]. All repair data sets laid over chromatin states and epigenetic factor sites are the merged sets of all time points of the two experiments.

**Expression data processing**. The expression data for the core clock genes (Supplementary Fig. 11a) were retrieved from the "Diurnal Database [http://diurnal. mocklerlab.org]" using the "long day" data set. The circadian RNA-seq data sets (used in Supplementary Fig. 11b,c) were retrieved from 'Sequence Read Archive (SRA) [https://www.ncbi.nlm.nih.gov/sra]'. The SRA accession numbers were obtained from Gene Expression Omnibus database (GSE43865)[40]. The raw sequence data sets were retrieved using the SRA toolkit and aligned with the reference genome (TAIR10) using Tophat[41] with default options. The genic FPKM values were computed as in XR-seq data analysis with the exception that strand separation was not applied. The three sets of RNA-seq samples collected at six different circadian time points (ZT) were used to compute rhythmicity by following the same procedure in XR-seq rhythmicity analysis. To compare TS XR-seq with RNA-seq phase values, we retrieved the significantly rhythmic genes that are common in both data sets (XR-seq TS and RNA-seq) by applying the cutoffs of BH.Q < 0.05 and Amplitude >1. 1754 genes commonly found as rhythmic in both data sets were used to generate the scatter plot and the histogram of the phase value differences.

**Repair oscillation and functional phase set enrichment**. The oscillation queries of the genomic bins and genes were performed using Metacycle software[42] with the rhythmic signal detection methods ARS (ARSER), JTK (JTK_CYCLE) and LS (Lomb-Scargle). For genomic bins, we used JTK and LS methods and applied 0.05 p-value cutoff. For genic TS and NTS oscillation detection, we applied the three methods and used stricter criteria: meta2d_BH.Q (false discovery rate based on the

integrated *p* values) <0.05 and meta2d_AMP (the amplitude) >1. We used PSEA tool[43] to cluster phase values in a given biological cluster. Functional annotations were downloaded from TAIR database[44] and reformatted to prepare for PSEA input. Genes with significantly oscillating TS repair were selected more strictly: meta2d_BH.Q < 0.05 and meta2d_AMP > 2 . We ran PSEA with the *q* and *p* value cutoffs (0.05) applied distinctly and combined the results.

**Data availability**. All sequencing data that support the findings of this study have been deposited in the National Center for Biotechnology Information Gene Expression Omnibus (GEO) and are accessible through the GEO Series accession number "GSE108932". All other relevant data are available from the corresponding authors on request.

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

## Acknowledgements

We thank Drs. Gregory P. Copenhaver and Jeffery Dangl for their help and comments on the manuscript. Drs. Eui Hwan Chung and Farid El Kasmi (Dangl lab) generously provided materials. This work was supported by National Institutes of Health projects ES027255 and GM118102. We wish to dedicate this paper to Professor Winslow Briggs on the occasion of his 90th birthday.

## Author contributions

O.O., A.S. and O.A. designed the study. O.O. and C.S.P. performed the experiments. O.O., A.S. and O.A. analyzed the data. All authors contributed to writing and reviewing the manuscript.

## Additional information

**Competing interests:** The authors declare no competing interests.

