## [Peer Review File · Nature Communications]

Reviewers' comments:

Reviewer #1 (Remarks to the Author):

Sancar paper-

The majority of UV-induced dimers, in the great majority of eukaryotes, are repaired by photolyases. However, if repair of dimers is observed in the absence of photoactivating light, NER-based repair of dimers can be employed as a proxy for NER in general, allowing us to draw some general conclusions about NER. One of the more important observations- made several decades ago- is that NER detects DNA damage via two distinct mechanisms, a global mechanism, and a more rapid form of repair that actually detects the stalling of RNA polymerase. Thus transcribed strands are repaired more rapidly than nontranscribed strands, as transcribed strands they are detected by both processes. This "transcription coupled repair" occurs in all eukaryotes, including plants. Plants carry homologs of the genes required for this process, and a single publication assaying repair in the transcribed vs nontranscribed strands of a single gene in Arabidopsis was published a few years ago.

Here the authors use a more recent, powerful, and global assay, already employed for other eukaryotes, to perform whole-genome analysis of repair of CPDs in Arabidopsis seedlings, and present evidence that transcribed strands are, in general, repaired more rapidly than nontranscribed strands in Arabidopsis (on transcribed regions are also investigated. More specifically, they take a snapshot of the frequencies of CPD-containing mapped oligonucleotides at 1/2 hr after irradiation of plants (in the dark) and find more oligos derived from transcribed strands (TS) than nonTS). This snapshot doesn't really establish a rate of repair, but if we assume that 1/2 hr after irradiation is an early time point (the authors should justify this, and I'm sure they can), then this result certainly indicates that NER is indeed more efficient, in plants as in other eukaryotes, and for plant genes in general, just as one would expect. Its much more thorough than the earlier published report.

The authors then go on to assay repair 1/2 hr after irradiation at various times during the day, in the hope of determining whether excision repair itself is subject to circadian regulation. Several very basic things are wrong with this analysis and so the results are unconvincing.

1) Circadian regulation refers to an internal clock, which is established by earlier solar training. In other words, a process is circadian if it continues to fluctuate in a rhythmic 24 hr pattern in the absence of light (or the continuous presence of light). This allows this investigator to distinguish between processes that are regulated by light (in plants- many) vs processes that are regulated by an internal clock. If the authors want to establish that repair itself is circadian-regulated, they need to perform a free- running experiment under constant (nonphotoreactivating) light (or dark), and show that there are peaks and troughs in repair- independent of the target sequence's transcription level.

2) The fact that genes whose transcription is known to be circadian are repaired (on the transcribed strand only) in a circadian pattern does NOT suggest that repair is regulated by a circadian clock - it probably simply reflects regular changes in transcription rate and therefore another instance of good old fashioned TCR. I was very surprised to see that the authors did not present data correlating transcription rates at various times of day with oligo frequencies (unless this was in the incomprehensible figure 3e?) for the highest-amplitude circadian-regulated genes, but instead present only the repair rates.

This paper needs to be longer and more thoughtful and perhaps is not suited to this format.

Sub-points:

I. 39-A minor point- I was a surprised to read that plants defective in excision repair were UV sensitive, given the presence of 2 photolyases Arabidopsis- perhaps this should be tempered by the statement "particularly in the absence of photoreactivating light?". In the Biedermeier reference (similarly Jiang get al 1997, Liu et al 2000) the plants recover in the dark. The Molinier paper doesn't

mention a dark recovery period, though they measure root growth only 24 hrs after irradiation, reflecting cell expansion rather than cell division.

"Efficiency" of repair- Normally efficiency would simply mean rate- but this is simply a snapshot of the frequency of various excised oligos at 30' after irradiation- no rate is determined. The authors need to tell us why this time point was chosen. The frequency of oligos from rapidly-repaired regions will indeed peak in concentration earlier than oligos from slowly repaired regions. But because the authors are only taking a snapshot they cannot determine the time of the peaks. The relative values of oligos from slowly vs rapidly repaired regions will shift continuously, and the frequency of oligos from slowly repaired regions may actually exceed that of oligos rapidly repaired regions at later time points (as the oligos are presumably short-lived and repair of "rapid regions" may be complete). I'm not at all suggesting that multiple time points be assayed- I just want to know why 30' was chosen. A discussion the rate of overall repair dimers in the absence of photoreactivation would probably nail this issue down.

I. 90 "...however this correlation may not be linear..." there is no way the reader can determine whether the correlation of repair rate with expression is linear given that the relative levels of expression of the genes in each quartile is not described (is the bottom quartile expressed at 1/1000th the level of the top quartile?) and, as discussed above, the rate of repair- even the relative rates of repair- is not determined here. Again, the relative concentrations of oligos from sequences with various rates of repair will vary continuously with time.

I. 103: the authors suggest that 3' ends of genes are repaired "less efficiently" due to abortive transcription induced by upstream CPDs. What is the density of induction of CPDs in this experiment, and (therefore) what fraction of genes would be expected to carry more than 1 CPD?

I 121 The authors assure us that the same general patterns of chromatin states and gene expression exist in all Arabidopsis samples, regardless of growth condition, tissue harvested, or stage of maturity. Can they provide some references to support this remarkable statement?

I 98: I believe the "9 chromatin states" defined in the reference provided describe histone codes, not the predicted function of a particular sequence (promoter, coding region etc). Thus this term applies to fig 2b, not 2a. Figure 2a needs some sort of legend that better defines each category. 5' end of gene =...what? Is it the first 50% of the transcribed region?

Reviewer #2 (Remarks to the Author):

This is review of the manuscript „Circadian clock- and transcription controlled genome-wide excision repair in Arabidopsis" submitted by Oztas et al. to Nature Communications. In this study, the authors analyzed dynamics of repair of UV-induced damage and showed that under the applied conditions there is strong preference for repair of transcribed strand by presumably transcription coupled repair (TCR) of the nucleotide excision repair pathway. TCR repaired genes are more frequently residing in euchromatin and are depleted of the silencing chromatin marks. Finally, TCR is strongly coupled to the circadian rhythms and genes involved in the same pathways are repaired synchronously.

This is timely study, with carefully conducted experiments. The experimental conditions were not natural, but in my opinion fully justified by the aim to analyze kinetics of the TCR in Arabidopsis and the necessity to suppressed photoreactivation. However, the authors need to adjust some of their statements (see below). The most exciting finding is the strong connection of TCR to the circadian clock, which is novel and will gain lots of attention in the Arabidopsis field.

Here are specific comments:

Abstract, line 11: The authors state „They remove UV-induced DNA lesions, and presumably other bulky DNA adducts, by nucleotide excision repair to maintain their genome integrity and their fitness". I find this oversimplification as plants employ also photoreactivation (direct reversal) to remove UV -

induced damage as proved by many studies.

The authors wrote „UV component of sunlight“ at several positions. Light is usually visible. Since UV-B and C are invisible for human eyes, it would be more accurate to write „UV component of solar radiation“.

Results p 3: The authors refer to SFs 3 and 4, which contain many timepoints. However, this is not introduced in the text. I find this confusing. In addition, it is not clear to me from the current description when the UV treatment was done for e.g. Z8, Z14, Z17 etc...

GRO-seq is not described in the materials section.

p 5 line 96: The authors state „We conclude that TCR dictates the genome-wide excision repair profile in Arabidopsis.“ This is true under the applied conditions. However, the conditions were not natural in terms of UV composition (unfiltered UV-C lamp) and irradiation regime (plants were placed to dark directly after the UV treatment, which practically never happens in nature). Therefore, I suggest that the authors weaken their statement.

p 6, line 115: As expected, DNA methylation ... caused severe depression in the repair of both strands because these marks are associated with heterochromatin“. In plants, DNA methylation occurs in three functional contexts CG, CHG and CHH, where H is A,T or C. While presence of all three contexts has silencing effects and occurs in heterochromatin, occurrence of CG methylation in gene bodies seems to even promote transcription (or at least not repress it). It would be interesting to see differences in TCR in genes containing gene body DNA methylation versus genomic regions labeled by all types of DNA methylation.

Discussion, lines 181-184: The authors discuss that there is less repair in heterochromatic compared to euchromatic regions. Recent study (ref 2) showed increased mutation rates of DNA methylated cytosines in heterochromatin compared to euchromatic regions. It would be good to discuss this as it nicely supports the author's model.

p 18, line 294: better „accession“ than „ecotype“

p 18, line 298: the times given here differ from those in the figures and supplementary figures.

p 18, line 300: Please specify the duration of irradiation.

Reviewer #3 (Remarks to the Author):

Review: Circadian clock and transcription-controlled genome-wide excision repair in arabidopsis

In this manuscript by Oztas et al., a study of genome-wide excision repair is conducted over a 24 hour period. The method used is XR-seq in which excised DNA fragments containing cyclopyrimidine dimers are isolated and sequenced. Analyses of the dataset are conducted including quantifying strand specific repair in active and inactive transcriptional regions of the genome, in regions close to various epigenetic marks, and in genes with rhythmic expression patterns controlled by the circadian clock. The authors conclude that the highest repair rates occur in actively transcribed regions of the genome.

The manuscript provides a valuable dataset examining the sites and rates of repair in the Arabidopsis

genome across a day. This can be used to compare against other genome wide datasets making it a valuable resource for those performing genome-wide studies in Arabidopsis. Furthermore, DNA repair mechanisms in plants have some differences compared to other organisms making this study important for comparing the functions of repair mechanisms across organisms.

The manuscript is presented in a clear and concise manner. The data is also presented clearly. This made for an enjoyable read.

There are no major changes that need to be made prior to publication.

In sum, this manuscript provides some novel insight into how plant DNA repair is controlled and links DNA damage repair to sites of active transcription. While this may not be surprising, the clearly presented data will be valuable to multiple research communities.

Reviewer #3 (Remarks to the Author):

Review: Circadian clock and transcription-controlled genome-wide excision repair in arabidopsis

In this manuscript by Oztas et al., a study of genome-wide excision repair is conducted over a 24 hour period. The method used is XR-seq in which excised DNA fragments containing cyclopyrimidine dimers are isolated and sequenced. Analyses of the dataset are conducted including quantifying strand specific repair in active and inactive transcriptional regions of the genome, in regions close to various epigenetic marks, and in genes with rhythmic expression patterns controlled by the circadian clock. The authors conclude that the highest repair rates occur in actively transcribed regions of the genome.

The manuscript provides a valuable dataset examining the sites and rates of repair in the Arabidopsis genome across a day. This can be used to compare against other genome wide datasets making it a valuable resource for those performing genome-wide studies in Arabidopsis. Furthermore, DNA repair mechanisms in plants have some differences compared to other organisms making this study important for comparing the functions of repair mechanisms across organisms.

The manuscript is presented in a clear and concise manner. The data is also presented clearly. This made for an enjoyable read.

There are no major changes that need to be made prior to publication.

In sum, this manuscript provides some novel insight into how plant DNA repair is controlled and links DNA damage repair to sites of active transcription. While this may not be surprising, the clearly presented data will be valuable to multiple research communities.

We are very glad that this reviewer found the manuscript and data to be clearly presented and the paper enjoyable to read.

Reviewer #2 (Remarks to the Author):

This is review of the manuscript „Circadian clock- and transcription controlled genome-wide excision repair in Arabidopsis“ submitted by Oztas et al. to Nature Communications. In this study, the authors analyzed dynamics of repair of UV-induced damage and showed that under the applied conditions there is strong preference for repair of transcribed strand by presumably transcription coupled repair (TCR) of the nucleotide excision repair pathway. TCR repaired genes are more frequently residing in euchromatin and are depleted of the silencing chromatin marks. Finally, TCR is strongly coupled to the circadian rhythms and genes involved in the same pathways are repaired synchronously.

This is timely study, with carefully conducted experiments. The experimental conditions were not natural, but in my opinion fully justified by the aim to analyze kinetics of the TCR in Arabidopsis and the necessity to suppressed photoreactivation. However, the authors need to adjust some of their statements (see below). The most exciting finding is the strong connection of TCR to the circadian clock, which is novel and will gain lots of attention in the Arabidopsis field.

Here are specific comments:

Abstract, line 11: The authors state „They remove UV-induced DNA lesions, and presumably other bulky DNA adducts, by nucleotide excision repair to maintain their genome integrity and their fitness “. I find this

oversimplification as plants employ also photoreactivation (direct reversal) to remove UV-induced damage as proved by many studies.

The statement in the abstract has been changed to be consistent with the fact that excision repair is not the sole repair mechanism for UV photoproducts: “They employ nucleotide excision repair to remove DNA-bulky adducts and to help eliminate UV-induced DNA lesions, so as to maintain their genome integrity and their fitness.”

The authors wrote “UV component of sunlight” at several positions. Light is usually visible. Since UV-B and C are invisible for human eyes, it would be more accurate to write „UV component of solar radiation“. 0

While UV is in fact visible to some organisms, we agree that “solar radiation” is a proper term and “solar radiation” has been incorporated in place of “sunlight”.

Results p 3: The authors refer to SFs 3 and 4, which contain many timepoints. However, this is not introduced in the text. I find this confusing. In addition, it is not clear to me from the current description when the UV treatment was done for e.g. Z8, Z14, Z17 etc...

Originally we referred to Supplemental Figure 4. This was erroneous, the corrected reference is Supplemental Figure 4b. Clearly the reviewer was confused by perusing the irrelevant material and many time points in Supplemental Figure 4a. This section has been re-written for clarity by adding relevant descriptive information in the text as well as the Legends to the Supplemental Figs. 3 and 4.

In Supplementary Figures 3 and 4, we showed the read length distribution of XR-seq samples and nucleotide content of the excised oligomers at different circadian timepoints. ZT2-ZT23 stand for the circadian timepoints. To maintain the flow of the text and not to confuse the readers, we briefly mention about the time points as “(ZT2-ZT23)” on p. 3 of the text and note to the reader that the time points are discussed later in the text.

GRO-seq is not described in the materials section.

We did not perform GRO-seq experiment. In our analysis, we used the publicly available dataset from Hetzel et al.,2016 (Reference 16). We added “previously reported” statement in the main text to clarify.

p 5 line 96: The authors state „We conclude that TCR dictates the genome-wide excision repair profile in Arabidopsis.“ This is true under the applied conditions. However, the conditions were not natural in terms of UV composition (unfiltered UV-C lamp) and irradiation regime (plants were placed to dark directly after the UV treatment, which practically never happens in nature). Therefore, I suggest that the authors weaken their statement.

We have appended the conclusion with” ...under our experimental conditions.”

p 6, line 115: As expected, DNA methylation ... caused severe depression in the repair of both strands because these marks are associated with heterochromatin“. In plants, DNA methylation occurs in three functional contexts CG, CHG and CHH, where H is A,T or C. While presence of all three contexts has silencing effects and occurs in heterochromatin, occurrence of CG methylation in gene bodies seems to even promote transcription (or at least not repress it). It would be interesting to see differences in TCR in genes containing gene body DNA methylation versus genomic regions labeled by all types of DNA methylation.

We agree that this would be an interesting comparison but it is impractical since genomic regions labeled by all types of DNA methylation do not undergo TCR.

Discussion, lines 181-184: The authors discuss that there is less repair in heterochromatic compared to euchromatic regions. Recent study (ref 2) showed increased mutation rates of DNA methylated cytosines in heterochromatin compared to euchromatic regions. It would be good to discuss this as it nicely supports the author's model.

We appreciate the reviewer's suggestion. We added "Our result is consistent with the observation that UV causes higher mutation rates of DNA methylated cytosines in heterochromatic regions than euchromatic regions"

p 18, line 294: better „accession“ than „ecotype“

We changed "ecotype" to "accession".

p 18, line 298: the times given here differ from those in the figures and supplementary figures.

We appreciate the reviewer finding this error which we have corrected.

p 18, line 300: Please specify the duration of irradiation.

We changed the description of UV irradiation to "10-days old seedlings were irradiated with 1 J/(m²s) UVC (254 nm) for 2 min (120 J/m² UVC)" in the materials section.

Reviewer #1 (Remarks to the Author):

Sancar paper-

The majority of UV-induced dimers, in the great majority of eukaryotes, are repaired by photolyases.

Photolyases appear to be common among plants but occurrence is spotty among other eukaryotes and many eukaryotes (e.g., mammals) have neither. Photolyases vary in their efficiency depending upon the particular enzyme and reaction condition. Excision repair also varies in efficiency even within the genome of individual cells as we report here.

However, if repair of dimers is observed in the absence of photoactivating light, NER-based repair of dimers can be employed as a proxy for NER in general, allowing us to draw some general conclusions about NER. One of the more important observations- made several decades ago- is that NER detects DNA damage via two distinct mechanisms, a global mechanism, and a more rapid form of repair that actually detects the stalling of RNA polymerase. Thus transcribed strands are repaired more rapidly than nontranscribed strands, as transcribed strands they are detected by both processes. This "transcription coupled repair" occurs in all eukaryotes, including plants. Plants carry homologs of the genes required for this process, and a single publication assaying repair in the transcribed vs nontranscribed strands of a single gene in Arabidopsis was published a few years ago.

Here the authors use a more recent, powerful, and global assay, already employed for other eukaryotes, to perform whole-genome analysis of repair of CPDs in Arabidopsis seedlings, and present evidence that transcribed strands are, in general, repaired more rapidly than nontranscribed strands in Arabidopsis (on transcribed regions are also investigated. More specifically, they take a snapshot of the frequencies of CPD-containing mapped oligonucleotides at ½ hr after irradiation of plants (in the dark) and find more oligos derived from transcribed strands (TS) than nonTS). This snapshot doesn't really establish a rate of repair, but if we assume that 1/2 hr after irradiation is an early time point (the authors should justify this, and I'm sure they can), then this result certainly indicates that NER is indeed more efficient, in plants as in other eukaryotes, and for plant genes in general, just as one would expect.

We reported earlier (Canturk et al (2016) PNAS 113, 4706-10) that in the absence of photoreactivating light, removal of CPDs from cultured plant cells as measured by slot blot is 'slow', for example, it takes almost 24 hours to remove 50 % of CPDs from the genome. In addition, excision assay products from mammalian cells were found to increase in abundance up to four hours (Choi et al (2014) NAR 42, e29). We state in the revised first paragraph of results that "This is a relatively early time point in the CPD repair time course which takes hours to complete".

Its much more thorough than the earlier published report.

The authors then go on to assay repair ½ hr after irradiation at various times during the day, in the hope of determining whether excision repair itself is subject to circadian regulation. Several very basic things are wrong with this analysis and so the results are unconvincing.

The "Several very basic things" listed below are responded on a point by point basis.

The reviewer misses a key point of this paper: This is the first repair paper genome-wide carried out on a whole multicellular organism.

1) Circadian regulation refers to an internal clock, which is established by earlier solar training. In other words, a process is circadian if it continues to fluctuate in a rhythmic 24 hr pattern in the absence of light (or the continuous presence of light). This allows this investigator to distinguish between processes that are regulated by light (in plants- many) vs processes that are regulated by an internal clock. If the authors want to establish that repair itself is circadian-regulated, they need to perform a free- running experiment under constant (nonphotoreactivating) light (or dark), and show that there are peaks and troughs in repair- independent of the target sequence's transcription level.

We appreciate the value of experiments done under constant environmental conditions (free-running) in revealing the innate clock, and we have performed such experiments in the past with mice. The reviewer suggests that such a free-running experiment is needed in our plant studies, but at the same time points to an inherent difficulty, that is, the dualing complications of either omitting light which will cause decline in plant health, or including constant light and dealing with possibly variable photoreactivation. We chose to use LD conditions to obtain data that is not confounded by these ambiguities. This regimen is commonly used in the circadian field [see Menet & Rosbash (2017 Nobel for circadian clock) eLIFE 2012. DOI:10.7554] who measured gene expression profile under LD conditions and they referred to the RNA expression profile "circadian oscillation". Thus, our data obtained under environmentally relevant conditions is in accord with the rather common practice in the circadian field.

To ascertain a link between repair and the clock we focused our data analysis in several instances on repair of core clock genes and compared the rhythms of repair with the known rhythmic patterns of gene expression for the corresponding genes. This comparison is first made with CCA1 in Fig. 3a, and the repair and transcription oscillations do in fact coincide as described in the text. The text also describes from Fig. 3d and e how numerous other clock genes exhibit peak repair at various times throughout the day, nevertheless, in each case repair rhythmicity parallels transcription rhythmicity.

It seems the reviewer fails to recognize this approach to establish circadian regulation of repair or for that matter fails to bring up any other meaningful approach be it genetic, biochemical or other. We acknowledge that diurnal variations such as light may demonstrate apparent circadian rhythmicity and contribute to the oscillations that we observe, and this is stated in the revised text with the example of photosynthetic genes.

Furthermore, the argument that the experiment was done under LD conditions and therefore it cannot be considered to show circadian control is incorrect. If as the reviewer claims that these are "processes that are

regulated by light” then the gene expression pattern in the light phase or the dark phase for that matter would not exhibit the unique patterns we observe. The reviewer’s argument would be tenable if when light is turned on a set of genes would synchronously turn on and with lights off another set of genes would synchronously turn on. What we observe is a circadian pattern of repair for the entire repairome over every phase of the circadian cycle (Fig. 3b). It is for the same reason that many Arabidopsis circadian clock experiments have been done under LD cycles.

We expect that by performing the experiment in free-running condition we might see a change in the amplitude of TCR repair or in the number of genes that have a circadian repair oscillation. However, this does not change the main conclusion that there are genes that show circadian oscillation of repair in their transcribed strands. Moreover, we showed that the TS repair in core-clock genes exhibits a strong and significant rhythmicity, and their phase values are very well-correlated with their circadian oscillation of transcription levels in the literature (Figure 3a,3d,3e). We believe that this clearly shows the circadian oscillation of repair of genes.

2) The fact that genes whose transcription is known to be circadian are repaired (on the transcribed strand only) in a circadian pattern does NOT suggest that repair is regulated by a circadian clock- it probably simply reflects regular changes in transcription rate and therefore another instance of good old fashioned TCR. I was very surprised to see that the authors did not present data correlating transcription rates at various times of day with oligo frequencies (unless this was in the incomprehensible figure 3e?) for the highest-amplitude circadian-regulated genes, but instead present only the repair rates.

The x-axis of Figure 3e has been properly labeled in the revised manuscript to clarify this Figure.

As we mentioned in the discussion, while XRseq monitors only transcription, RNA-seq and microarray data capture posttranscriptional events in addition to transcriptional events. Therefore, comparison of XRseq data with these datasets are not informative. For this reason, in our analysis we used a GRO-seq dataset which shows transcription. Unfortunately, there is no available circadian GROseq dataset.

This paper needs to be longer and more thoughtful and perhaps is not suited to this format.

Our manuscript is related to many research areas: DNA repair, circadian clock, plant biology, bioinformatics, and epigenetics. To make our manuscript understandable and enjoyable for readers with different backgrounds, we wrote it clearly and simple, but, at the same time, scientifically robust. Notably, Reviewer 3 states “The manuscript is presented in a clear and concise manner. The data is also presented clearly. This made for an enjoyable read” and “In sum, this manuscript provides some novel insight into how plant DNA repair is controlled and links DNA damage repair to sites of active transcription”.

Sub-points:

I. 39-A minor point- I was surprised to read that plants defective in excision repair were UV sensitive, given the presence of 2 photolyases Arabidopsis- perhaps this should be tempered by the statement “particularly in the absence of photoreactivating light?”. In the Biedermeier reference (similarly Jiang et al 1997, Liu et al 2000) the plants recover in the dark. The Molinier paper doesn’t mention a dark recovery period, though they measure root growth only 24 hrs after irradiation, reflecting cell expansion rather than cell division.

Willing et al., (ref 2) reported higher mutation accumulation in *uvh1* mutant plants, which are defective in nucleotide excision repair, compared to WT under photoreactivating light. This suggests that plant photolyases have limited capacity, and nucleotide excision repair has an important function to remove UV-photoproducts even in the presence of photoreactivation. However, the reviewer is right that the referenced papers showed the excision repair mutation effect at dark. Therefore, we also showed Willing et al. as a reference for that statement.

“Efficiency” of repair- Normally efficiency would simply mean rate- but this is simply a snapshot of the frequency of various excised oligos at 30’ after irradiation- no rate is determined. The authors need to tell us why this time point was chosen. The frequency of oligos from rapidly-repaired regions will indeed peak in concentration earlier than oligos from slowly repaired regions. But because the authors are only taking a snapshot they cannot determine the time of the peaks. The relative values of oligos from slowly vs rapidly repaired regions will shift continuously, and the frequency of oligos from slowly repaired regions may actually exceed that of oligos rapidly repaired regions at later time points (as the oligos are presumably short-lived and repair of “rapid regions” may be complete). I’m not at all suggesting that multiple time points be assayed- I just want to know why 30’ was chosen. A discussion the rate of overall repair dimers in the absence of photoreactivation would probably nail this issue down.

As described above, it takes almost 24 hours to remove 50% of the CPDs from plant cells in the absence of photoreactivating light. Thus repair at 30 minutes reflects an early time point that reflects the rate of repair. In addition, it is a practical time point because sufficient repair product is available for the XR-seq procedure and at later time points degradation becomes a greater factor. As noted above, we state in the revised manuscript, first paragraph of discussion that 30 minutes is a relatively early repair time point.

I. 90 “..however this correlation may not be linear...” there is no way the reader can determine whether the correlation of repair rate with expression is linear given that the relative levels of expression of the genes in each quartile is not described (is the bottom quartile expressed at 1/1000th the level of the top quartile?) and, as discussed above, the rate of repair- even the relative rates of repair- is not determined here. Again, the relative concentrations of oligos from sequences with various rates of repair will vary continuously with time.

Here, we agreed with the reviewer and replaced the fig S6 with the scatter plots showing the correlation between repair and expression in TS and NTS. We observed a positive correlation between TS XR-seq and GRO-seq while there was no strong correlation between NTS repair and GRO-seq. By applying LOESS curve fitting, we show that TS repair in the highly expressed genes is saturated. The corresponding sentence in the main text was modified.

I. 103: the authors suggest that 3’ ends of genes are repaired “less efficiently” due to abortive transcription induced by upstream CPDs. What is the density of induction of CPDs in this experiment, and (therefore) what fraction of genes would be expected to carry more than 1 CPD?

The dose used produced approximately 0.7 CPDs per kbp. The precise lesion distribution varies as a function of predetermined (frequency of dipyrimidines) and stochastic parameters. It is reasonable that in many cases there are multiple targets for TCR per gene. The occurrence of multiple lesions per gene is discussed in the revised text in the context of the less efficient repair at the 3’ end of genes.

I 121 The authors assure us that the same general patterns of chromatin states and gene expression exist in all Arabidopsis samples, regardless of growth condition, tissue harvested, or stage of maturity. Can they provide some references to support this remarkable statement?

In our analysis, we compared our XR-seq data with publicly available chromatin databases. Some of these datasets were obtained from Arabidopsis plants with different growth and developmental stages, which might not seem to be ideal. We agree with the reviewer that the sentence in the text ignores the chromatin plasticity. We modified the relevant text to clarify the rationale of why such an analysis is appropriate. We measure the effect of a chromatin state by applying a comparative approach. First, the cumulative effect of chromatin states is normalized by randomized regions. Second the effect of chromatin state is evaluated in the chromatin region (window) compared with its flanking regions. Even if there could be changes in the chromatin states between different experimental conditions, the regions that are constant across samples are sufficient enough to yield the signal we

captured. Based on the normalization processes we applied, we believe that our state-of-the-art analysis which is commonly applied by numerous genome-wide comparative studies is appropriate.

I 98: I believe the "9 chromatin states" defined in the reference provided describe histone codes, not the predicted function of a particular sequence (promoter, coding region etc). Thus this term applies to fig 2b, not 2a. Figure 2a needs some sort of legend that better defines each category. 5' end of gene =...what? Is it the first 50% of the transcribed region?

Reviewer is right about the definition of the 9 chromatin states which were derived from the combinatorial histone markers (ref 18). However, this term applies to fig 2a perfectly, as we measure repair in these 9 chromatin states. To clarify, we reworded the sentence in the main text. The nine chromatin states were primarily named as "state 1" to "9". The states were renamed based on the genomic regions enriched with these states by the same author (ref 36). The annotations do not necessarily indicate the predicted functions; however, they are mostly associated with them. Their lengths and borders are also not precise (averaged genome was arbitrarily divided into 200bp bins to define a chromatin state unit). We believe that the annotations of the chromatin states make the figure easier to interpret and it is appropriate to leave the detailed investigation of the chromatin state annotations to the reader provided with the necessary references.

REVIEWERS' COMMENTS:

Reviewer #1 (Remarks to the Author):

I sincerely apologize to the authors for a seriously garbled passage in my review of their previous submission!

Again, this is a nice paper that assays the sequence and strand-specificity of NER in plants. This strand-specificity has been published previously for a single plant gene, and this paper convincingly demonstrates that TCR in Arabidopsis occurs throughout the genome, as in other eukaryotes. In their rebuttal the authors protested that I didn't appreciate the novelty of performing this study in a multicellular organism. That's true, I don't think that the multicellularity adds much, conceptually.

I continue to take issue with the author's second point: that repair itself is clock-controlled (see title).

The authors demonstrate- or tell us (see below)- that repair of genes whose transcription is subject to circadian rhythms occurs in a circadian rhythm. They conclude from this that repair is "regulated" by a circadian clock. The circadian rhythm of repair of genes transcribed in a circadian pattern is the inevitable result of the coupling of repair and transcription. I feel that it is inappropriate to describe this as "circadian regulation of repair".

Let me provide an example: If we treated the plants with a pathogen-derived protein- flagellin- then the plants would undergo a pathogen response, which includes changes in the transcription of certain genes. We'd expect to see the rate of repair of the transcribed strand increase in the flagellin-induced genes, but we wouldn't conclude- from this data- that repair is regulated by flagellins. Similarly, when the Hanawalt lab first observed TCR and found that its rate was further enhanced in the transcribed strand of a temperature-controlled gene at permissive temperature, they concluded that repair was transcription-coupled, not that repair was controlled by temperature.

If we are observing a circadian pattern in repair that inevitably results from TCR (as described in the example above), then all genes subject to circadian regulation of transcription would be subject to circadian variation in repair. The authors tell us that this is indeed what is observed for core circadian clock genes- that the "maximum repair phase of each clock gene coincides with its reported maximum transcription phase". Two requests here:

- 1) could this be clearly illustrated? Figure 3d and 3e both report the phase and amplitude of the repair differential across time for a few clock-regulated genes, but the reader is not provided with any data on the correlation between peak of repair rate and peak of transcript level. Where is the data on the correlation of transcription and repair that justifies the statement quoted above? For example, Figure 3d could simply have a line of another color reflecting the transcript level, together with another axis.
- 2) Please tell us whether the 4,000 or so genes described as exhibiting circadian repair also are subject to circadian regulation of transcription? If so, would it be possible to present a graph of repair phase vs transcript phase, to illustrate the degree of correlation?
- 3) I'm afraid that I still do not entirely understand Fig. 3e. The Y axis is "significance"- a log scale of P values- significance of what? I'm guessing of the fact that there is a circadian effect on repair?

L 171: In a revised sentence the authors state that "While these oscillations demonstrate circadian clock control of gene expression, in some cases, such as light-responsive photosynthetic genes, circadian expression is controlled directly by diurnal environmental exposures". Circadian effects are entrained by light, but I don't think the authors really are talking about entrainment here. I think they're saying that some circadian gene expression is clock-controlled but other types of "circadian" expression are controlled directly by light. Plant biologists especially need to distinguish between

entrained, clock regulated processes- which they (and I think other biologists) describe as circadian- and the many processes regulated more directly by light (which they describe as "light regulated"). We should not conflate the two concepts and describe both as "circadian".

Reviewer #2 (Remarks to the Author):

The authors have successfully addressed all my comments, except for one, where they probably did not understand my point.

Originally I wrote: p 6, line 115: As expected, DNA methylation ... caused severe depression in the repair of both strands because these marks are associated with heterochromatin". In plants, DNA methylation occurs in three functional contexts CG, CHG and CHH, where H is A,T or C. While presence of all three contexts has silencing effects and occurs in heterochromatin, occurrence of CG methylation in gene bodies seems to even promote transcription (or at least not repress it). It would be interesting to see differences in TCR in genes containing gene body DNA methylation versus genomic regions labeled by all types of DNA methylation.

The authors replied: We agree that this would be an interesting comparison but it is impractical since genomic regions labeled by all types of DNA methylation do not undergo TCR.

I understand this and agree with the authors. However, the critical difference lies in genes as some are fully unmethylated, while other carry CG DNA methylation in Arabidopsis. Hence, if the authors are correct, DNA methylated genes (CG only) should not be processed by TCR. More likely, CG DNA methylation does not interfere with TCR and it is heterochromatic nature of TE loci what blocks TCR. The authors can easily test this hypothesis as the "gene body methylated genes" are known in Arabidopsis.

We thank the two reviewers who have read and commented on our revised manuscript. Here are our point-by-point responses to the specific points.

I sincerely apologize to the authors for a seriously garbled passage in my review of their previous submission!

Again, this is a nice paper that assays the sequence and strand-specificity of NER in plants. This strand-specificity has been published previously for a single plant gene, and this paper convincingly demonstrates that TCR in Arabidopsis occurs throughout the genome, as in other eukaryotes. In their rebuttal the authors protested that I didn't appreciate the novelty of performing this study in a multicellular organism. That's true, I don't think that the multicellularity adds much, conceptually.

Reviewer #1 remains unconvinced that performing transcription-coupled repair experiments on a whole multicellular organism for the first time "adds much" to the subject, while we believe that it is a breakthrough. To understand this achievement, it is important to consider that circadian rhythms are not coordinated and sustained in cells in culture, and our study could not have been done with cultured cells. Furthermore, the methods we developed measure repair genome-wide, and provide very high quality information about ongoing transcription, genome-wide. Our methods and findings promote inquiry and open the door to investigation into areas such as tissue-specific gene expression and repair, repair pathways operative under real-world conditions, and efforts to improve plant performance.

I continue to take issue with the author's second point: that repair itself is clock-controlled (see title).

The authors demonstrate- or tell us (see below)- that repair of genes whose transcription is subject to circadian rhythms occurs in a circadian rhythm. They conclude from this that repair is "regulated" by a circadian clock. The circadian rhythm of repair of genes transcribed in a circadian pattern is the inevitable result of the coupling of repair and transcription. I feel that it is inappropriate to describe this as "circadian regulation of repair".

Let me provide an example: If we treated the plants with a pathogen-derived protein- flagellin- then the plants would undergo a pathogen response, which includes changes in the transcription of certain genes. We'd expect to see the rate of repair of the transcribed strand increase in the flagellin-induced genes, but we wouldn't conclude- from this data- that repair is regulated by flagellins. Similarly, when the Hanawalt lab first observed TCR and found that its rate was further enhanced in the transcribed strand of a temperature-controlled gene at permissive temperature, they concluded that repair was transcription-coupled, not that repair was controlled by temperature.

The reviewer states that we claim "that repair itself is clock-controlled". In fact, we claim no such thing but we explicitly state the opposite. We make an explicit point that whereas in mammalian cells the basal (global) repair activity is controlled by the circadian clock which regulates the expression of the rate-limiting protein XPA, Arabidopsis does not have an XPA homolog and the excision repair activity per se does not show a circadian pattern. What we show is that because transcription of many genes in Arabidopsis is controlled by the clock, and

because we find a very strong coupling of transcription to excision repair of the transcribed strand, repair of clock-controlled genes exhibits a strong circadian pattern.

The reviewer then states that if gene transcription is induced by a genotoxicant or temperature one wouldn't call it genotoxicant or temperature-controlled repair and therefore we should not refer to the transcription-coupled repair we observe as circadian. There is an important difference between the example he/she cites and what we observe. The cited examples are not rhythmic, the coupled repair we observe is rhythmic. If it is rhythmic because the transcription is rhythmic can one call it circadian or not is a semantic not a scientific question. As an example, in a typical presentation by circadian researchers on the effect of the circadian clock on human health, usually it is stated that the circadian clock controls blood pressure, and it is stated that blood pressure exhibits rhythmicity. Now, the core clock genes control the genes that control blood pressure and when one says that blood pressure is clock controlled it is implicit that the clock controls genes that affect blood pressure in a rhythmic manner. Again, this is a matter of semantics and we have chosen to use the conventional designation in the circadian field.

If we are observing a circadian pattern in repair that inevitably results from TCR (as described in the example above), then all genes subject to circadian regulation of transcription would be subject to circadian variation in repair. The authors tell us that this is indeed what is observed for core circadian clock genes- that the “maximum repair phase of each clock gene coincides with its reported maximum transcription phase”. Two requests here:

1) could this be clearly illustrated? Figure 3d and 3e both report the phase and amplitude of the repair differential across time for a few clock-regulated genes, but the reader is not provided with any data on the correlation between peak of repair rate and peak of transcript level. Where is the data on the correlation of transcription and repair that justifies the statement quoted above? For example, Figure 3d could simply have a line of another color reflecting the transcript level, together with another axis.

2) Please tell us whether the 4,000 or so genes described as exhibiting circadian repair also are subject to circadian regulation of transcription? If so, would it be possible to present a graph of repair phase vs transcript phase, to illustrate the degree of correlation?

We do in fact observe positive associations between transcription and repair among the Arabidopsis clock genes, as transcription of these genes is available from the literature. In the revised manuscript, we have included a new supplemental Fig. 11 that recapitulates, as suggested by the reviewer, Fig. 3d and shows, for each gene, the transcription and repair profiles on the same plot. Overall the profiles correspond closely. We also measured the correlations of phase values derived from XR-seq (transcribed strand) and RNA-seq under long day conditions. The unsurprising result is that the phase values overall correlate well with each other.

3) I'm afraid that I still do not entirely understand Fig. 3e. The Y axis is “significance”- a log scale of P values- significance of what? I'm guessing of the fact that there is a circadian effect on repair?

In the revised Legend to Fig. 3e we have restated the description to improve clarity: “Phase value (x axis) vs $-\log p$ value (significance of rhythmicity, y-axis) ...”. This type of plot is not novel to this paper.

L 171: In a revised sentence the authors state that “While these oscillations demonstrate circadian clock control of gene expression, in some cases, such as light-responsive photosynthetic genes, circadian expression is controlled directly by diurnal environmental exposures”. Circadian effects are entrained by light, but I don’t think the authors really are talking about entrainment here. I think they’re saying that some circadian gene expression is clock-controlled but other types of “circadian” expression are controlled directly by light. Plant biologists especially need to distinguish between entrained, clock regulated processes- which they (and I think other biologists) describe as circadian- and the many processes regulated more directly by light (which they describe as “light regulated”). We should not conflate the two concepts and describe both as “circadian”.

We have used “rhythmic” rather than “circadian” as suggested.

The authors have successfully addressed all my comments, except for one, where they probably did not understand my point.

Originally I wrote: p 6, line 115: As expected, DNA methylation ... caused severe depression in the repair of both strands because these marks are associated with heterochromatin“. In plants, DNA methylation occurs in three functional contexts CG, CHG and CHH, where H is A,T or C. While presence of all three contexts has silencing effects and occurs in heterochromatin, occurrence of CG methylation in gene bodies seems to even promote transcription (or at least not repress it). It would be interesting to see differences in TCR in genes containing gene body DNA methylation versus genomic regions labeled by all types of DNA methylation.

The authors replied: We agree that this would be an interesting comparison but it is impractical since genomic regions labeled by all types of DNA methylation do not undergo TCR.

I understand this and agree with the authors. However, the critical difference lies in genes as some are fully unmethylated, while other carry CG DNA methylation in Arabidopsis. Hence, if the authors are correct, DNA methylated genes (CG only) should not be processed by TCR. More likely, CG DNA methylation does not interfere with TCR and it is heterochromatic nature of TE loci what blocks TCR. The authors can easily test this hypothesis as the "gene body methylated genes" are known in Arabidopsis.

We agree with the reviewer that the analysis of methylation on repair would be interesting. We would like to leave it to a follow-up study where we will measure the effect of different methylation type on not only repair but also damage formation.